# Accelerated corrosion of low carbon steel by oscillatory acidic streams generated with a bio-inspired claw device

**Francisco A. Godínez**[1⊙], **Marvin Montoya-Rangel**[2⊙], **Rodrigo Montoya**[2⊙]*

**1** Instituto de Ingeniería, Unidad de Investigación y Tecnología Aplicadas, Universidad Nacional Autónoma de México, Apodaca, Nuevo León, México, **2** Facultad de Química, Unidad de Investigación y Tecnología Aplicadas, Universidad Nacional Autónoma de México, Apodaca, Nuevo León, México

⊙ These authors contributed equally to this work.
* rmontoyal@unam.mx

**Data Availability Statement:** The required DOI is: 10.5281/zenodo.10573937.

**Funding:** UNAM-DGAPA-PAPIIT IG100623 program has financed this work. MMR thanks

## Abstract

A mechanical device inspired by the pistol shrimp snapper claw was developed. This technology features a claw characterized by a periodic opening/closing motion, at a controlled frequency, capable of producing oscillating flows at transitional Reynolds numbers. An innovative method was also proposed for determining the corrosion rate of carbon steel samples under oscillating acidic streams (aqueous solution of HCl). By employing very-thin carbon steel specimens (25 $\mu m$ thickness), with one side coated with Zn and not exposed to the stream, it became possible to electrochemically sense the Zn surface once the steel sample was perforated, thus providing the average dissolution rate into the most relevant pit on the steel surface. Furthermore, a laser light positioned beneath the metallic sample, along with a camera programmed to periodically capture images of the steel surface, facilitated the accurate counting of the number of newly formed pits. The system consisting of the thin steel sample and the Zn coating can be seen as a type of corrosion sensor. Furthermore, the proposed laser illumination method allows corroborating the electrochemical detection of pits and also establishing their location. The techniques crafted in this study pave the way for developing alternative corrosion sensors that boast appealing attributes: affordability, compactness, and acceptable accuracy to detect in time and space localized damage.

## Introduction

In recent decades, the study of periodic flows (oscillatory and pulsatile) has attracted considerable interest among many scientists. This is primarily motivated by their potential applications in the environmental/coastal, biological, medical/health and industrial sectors [1]. Within this last category, where reciprocating flows play a predominant role, we can highlight applications such as oil well drilling [2], heat exchangers in Stirling motors, pulse-tube cryocoolers and gas turbines [3–5]. On the other hand, among the most notable advances in the medical/health field include artificial hearts and ventricular assist devices whose design involves the utilization of different types of pumps (diaphragm, pusher-plate, peristaltic, centrifugal and axial flow) to

UNAM-DGAPA for the postdoctoral fellowship throughout the program POSDOC.

**Competing interests:** The authors have declared that no competing interests exist.

produce pulsatile blood flow [6, 7]. Further significant developments in the medical area are the microfluidic systems in which oscillatory and electroosmotic pumps or pneumatic micro-pumps are used to produce pulsatile flows for mimicking physiological systems, to alter or enhance cell cultures and for the automation of bioassays [8]. In this research, a different and less conventional technique than the pumping technologies mentioned above was employed to produce oscillatory flows. The proposed technology consists of a mechanical device inspired by the snapping-claw mechanism of pistol shrimps to create high-speed liquid jets [9]. The first version of the device was designed to be manually triggered to produce the sudden closure (by the contraction of rubber bands) of a pivoting claw and, consequently, a jet of cavitating fluid at a time, this was useful for studying the synergy between corrosion/cavitation/erosion in metallic samples immersed in a saline electrolyte [10]. A redesign of the apparatus used in this work also includes a mobile claw but actuated with a four-bar mechanism driven in turn with an electric motor whose angular speed is adjustable, in this way it is possible to generate a reciprocating motion of the claw and thus an oscillatory flow together with the production of jets and the shedding of vortices at a controlled frequency. It was first thought to follow up on the research reported in [10] and consequently employ saline electrolytes to experimentally study the accelerated corrosion of low-carbon steel samples. Nevertheless, the highly complex nature of the oxide layers that form on the iron surface, particularly in corrosive environments containing NaCl [11–13], and the exceptional adherence of some of these layers to the surface, even withstanding oscillatory streams, led to the decision of eliminating this variable by employing an acidic environment. Acid solutions are commonly used in different areas of industry as pickling, cleaning, descaling, and oil-well acidizing. Acidic solutions are often employed to eliminate unwanted scale and rust in various industrial processes. Although acid solutions find widespread use in industry, they pose a significant challenge for steel due to its severe corrosion in acidic environments. During the acidizing stimulation process, hydrochloric, hydrofluoric, acetic, or formic acids are introduced into the well, leading to serious corrosion problems. Without corrosion inhibitors, the overall corrosion rate (CR) can be remarkably high ($\geq$100 mm/year) and may escalate exponentially with rising temperatures and acid concentrations [14]. Oil-well acidizing, for example, entails injecting acids into well systems, which are composed of steel tubes. The choice of acids used depends on the characteristics of the underground reservoir, and the treatment typically involves injecting acids at various concentrations. The most frequently utilized conventional acids are HCl, HF, acetic, and formic acids. However, numerous different combinations of these acids have also been employed in various scenarios [15]. The majority of acidizing treatments utilize HCl at concentrations ranging from 5% to 28% [16]. HCl offers an advantage over other mineral acids in acidizing operations because it forms metal chlorides, which exhibit high solubility in the aqueous phase [15]. Due to the aforementioned issues, the acidizing process necessitates a high level of corrosion protection for tubular materials and other equipment used. However, prior to implementing any protective measures, it is imperative to thoroughly assess the corrosion rate of the equipment under these chemical conditions. The experiments in this study aimed to investigate the accelerated corrosion of thin low-carbon steel specimens immersed in an oscillating acidic stream. The exposed surface of each specimen to the flow remained as bare steel, while the opposite side was coated with a zinc layer. The corrosion-erosion process was electrochemically monitored from the beginning of each test until the steel was perforated and the zinc was detected, providing the average dissolution rate in the most relevant pit on the metal surface. To complement the electrochemical measurements, a laser was employed to shine on the hidden face where the zinc was deposited, and a video camera was installed to continuously observe the surface of the sample exposed to the flow until the first light spot was identified, thus indicating the time and location of the material puncture. The electrochemical

monitoring and the illumination method yield pitting ocurrence times with a maximum difference of one hour, which is acceptable considering the stochastic and complex nature of the synergistic process between corrosion and erosion caused by flow. These findings promise a substantial leap forward in localized corrosion detection. They pave the way for the creation of sensors utilizing the interaction between two metals, introducing an innovative technology capable of efficiently detecting the impact of electrolyte flows in diverse regimes.

## Materials and methods

### Mechanical device

Fig 1 illustrates the mechanical apparatus used for accelerated corrosion tests. The cyclic opening and closing motion of the mobile claw is achieved by means of a conventional four-bar mechanism driven by an electric motor. The oscillating up and down motion of the output link (ol), indicated by the arrow (14), is transmitted to the mobile claw by means of the link (7). Of course, the opening and closing speeds of the mobile claw are controlled by the speed of rotation of the motor shaft, whose direction of rotation is shown by the arrow (15), and also by the arrangement and dimensions of the bars that make up the mechanism.

Fig 2a shows in its upper part a sequence of movements of the mobile claw; the starting position is when it is closed at $\theta = 5.4°$. The claw gradually opens counterclockwise until it reaches the maximum opening when $\theta = 34.3°$. After this point, the claw begins its closing motion in the clockwise direction, an intermediate position when it rotates in this orientation is $\theta = 22.1°$, in the final state the claw is closed reaching the same starting position when $\theta = 5.4°$, thus ending a complete cycle of motion. On the other hand, the continuous curve in blue represents the angular displacement as a function of time $\theta(t)$. This curve was obtained using the Tracker software [17] by analyzing a total of 3 videos of the angular motion of the claw for a motor rotation speed of about 150 rpm. The angular velocity curve $\omega(t)$ was obtained by numerically differentiating (using central differences) the curve representing $\theta(t)$. The angular velocities that characterize the motion of the claw with the proposed device reach maximum values in the order of 6 $rad/s$. This value contrasts sharply with the 180 $rad/s$ developed by a claw system designed to produce cavitating flows [18–20], which is used to study the damage in metallic samples due to synergistic effects between corrosion-erosion-cavitation [10]. Fig 2b

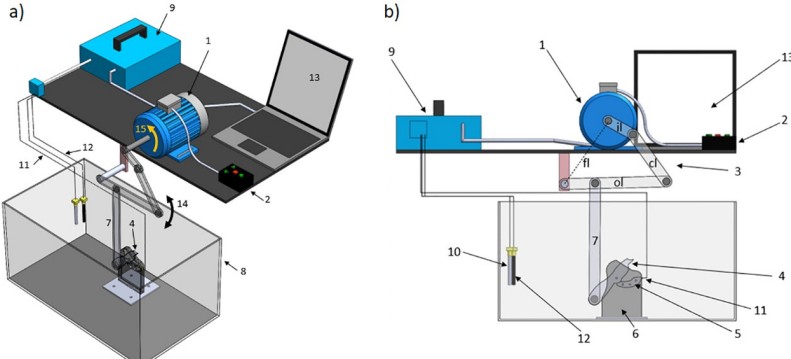

**Fig 1. Mechanical apparatus for accelerated corrosion testing.** a) Isometric view. b) Front view. 1) Electric motor, 2) motor angular speed control, 3) four-bar mechanism (composed of an input link (il), a coupler link (cl), a fixed link (fl) and an output link (ol)), 4) mobile claw, 5) fixed claw, 6) two plates to support the claws, 7) a link to transmit the oscillating motion of the output link (ol) to the mobile claw, 8) rectangular tank, 9) potentiostat, 10) reference electrode (RE), 11) working electrode (WE), 12) counter electrode (CE), 13) laptop, 14) this arrow indicates the directions of oscillating motion of the output link, 15) this arrow indicates the direction of rotation of the motor shaft.

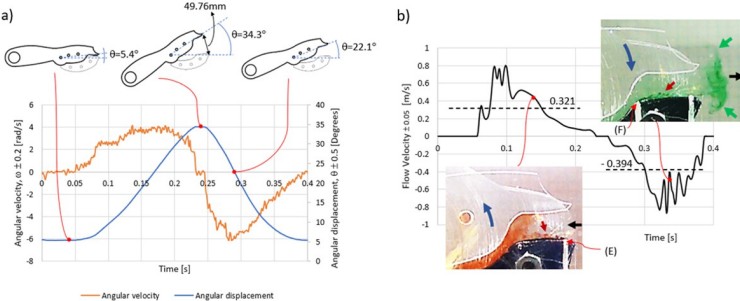

**Fig 2. Claw motion and induced flow.** a) Displacement and angular velocity developed by the moving claw for an angular velocity of the motor (and therefore of the bar (il)) of about 150 rpm. b) Oscillating flow velocities produced by the opening and closing motions of the claw. The blue arrows in each inset indicate the direction of claw rotation, the red arrows are used to note the streaklines produced and the black arrows indicate the direction of flow. The green arrows are used to indicate two counter-rotating vortices. The dashed lines represent the average value of the velocity useful for estimating the Reynolds numbers that characterize the flow. The standard deviation was calculated using three data runs.

depicts the flow velocity (induced by the mobile claw) versus time curve. This line was estimated from a video analysis by calculating the filling/emptying volume $V(t)$ of the socket formed by the fixed and mobile claws as well as the two supporting side plates, as shown in Fig 1 and S1A Fig in S1 Appendix. Also in the analysis it was necessary to calculate the volumetric flow $\dot{V}(t)$, as well as the area of the mouthpiece $A(t)$ through which the fluid is sucked and expelled, depending on whether the socket is filled or emptied, respectively. A more detailed explanation of the approach employed is given in S1 Appendix. The inset at the lower left of Fig 2b shows a snapshot of the opening motion of the claw as it moves counterclockwise as indicated by the arrow in blue. The red arrow points to streaklines drawn with red dye injected at port ($E$); clearly the flow develops to the left (black arrow) filling the back of the socket and the dye streaks show a wavy fluctuating type of motion suggesting the possibility of dealing with a transitional or even turbulent flow [21]. The average flow velocity during the filling of the socket is $0.321 m/s$ (represented by a dotted line, see Fig 2b), thus a Reynolds number of $Re \approx 1920$ can be estimated (assuming a characteristic length $\epsilon = 6mm$, please refer to S1 Appendix), which rules out a turbulent flow indicating a transient regime. On the other hand, the inset at the upper right of Fig 2b shows a snapshot of the flow characteristics as the socket volume empties while the claw closes in a clockwise direction, as indicated by the blue arrow. In this case, a green dye was released from port ($F$). Now the flow develops to the right (black arrow), and downstream vorticities (indicated by the green arrows) are observed just outside the mouthpiece, something similar has been reported with other claw systems [18, 19, 22]. Likewise, the streaklines show the typical fluctuations that suggest a transient or even turbulent flow regime. To discern between these two possibilities, the average flow velocity was also estimated giving a value of $0.394 m/s$ and thus a $Re \approx 2356$. This result indicates a transitional flow. It is important to note that the calculation of Reynolds numbers was made considering the properties of water under constant laboratory conditions, $T = 20°C$ and $P = 101.2$ kPa. This does not lead to considerable errors in our estimations since the solution used for the experiments was prepared with a low concentration of HCl, so the properties of the solution are very close to those of water [23]. The findings described above clearly show the capability of the mechanical device to produce oscillating flows defined by a period of $T = 0.4s$. In terms of a Strouhal number ($St$), which represents the ratio of inertial forces due to the local

acceleration of the flow to the inertial forces due to the convective acceleration, one has that [24, 25]

$$St = \frac{fA}{v} = 0.32 \tag{1}$$

where $A$ = is the amplitude of claw oscillation ($A \approx 50mm$, see Fig 2a), $f$ = claw oscillation frequency ($f = T^{-1} = 2.5Hz$), and $v$ = flow rate ($v = 0.394m/s$).

Also in the present context, $f$ represents the vortex shedding frequency. This detachment of vortical structures is a characteristic phenomenon at intermediate Strouhal numbers [25, 26] such as that estimated with Eq 1.

Another important dimensionless parameter to describe the oscillatory flow generated with the apparatus is the Womersley number ($Wo$), which compares the transient inertial effects to viscous forces. $Wo$ can be defined in terms of the $St$ and $Re$ numbers as [7]

$$Wo = \sqrt{\frac{ReSt}{2}} \approx 19 \tag{2}$$

this Womersley number value indicates that the oscillatory inertial force dominates the dynamics and strongly modifies the mean flow profile. In this regime, when the pressure gradient is reversed, it takes some time before the pressure gradient can change the direction of flow, leading to a phase change between the fluid flow and the pressure gradient.

It is worthy of note that the design of the mechanical apparatus allows working at much lower angular velocities to produce laminar flows. Nevertheless, it was decided to conduct the experiments in transient regimes, since the shear stresses on the exposed surface of the samples (and therefore the erosive-corrosive effects) are expected to be more accentuated than in laminar streams [27]. On the other hand, it is also important to mention that when the motor is operated at speeds above 250 rpm, mechanical vibrations are induced throughout the device that can hinder both electrochemical and optical sensing.

## Electrodeposition of zinc

The electrodeposition of zinc onto steel was conducted within a two-electrode cell, as illustrated in Fig 3. For the anode, a high-purity GoodFellow$^{TM}$ zinc wire (containing 99.99 wt. % Zn, 5 PPM Cd, 7 PPM Si, less than 1 PPM Cu, 5 PPM Mn, and 15 PPM Pb with 25 PPM Fe) was utilized. Meanwhile, a 25 $\mu$ m-thick AISI 1010 shim stock steel served as the cathode (substrate). Before the zinc plating process, the steel surface underwent polishing using 800-grit SiC emery paper, followed by a 15-second immersion in a 15 wt. % hydrochloric acid (HCl) solution at 25˚C. The deposition area was standardized at 1 cm$^2$, by masking the remaining area of one side of the electrode and the entirety of the opposite side with $3M^{TM}$ electroplating tape, with a 1.5 cm separation between the zinc electrode and the steel shim. The electrolyte comprised a 1:1 volume ratio of 15 wt. % HCl acid solution and deionized water. The total volume of electrolyte in the system was 60 cm$^3$, and the electrodeposition procedure was conducted within a 100 cm$^3$ glass beaker. Zinc was electrodeposited onto the steel surface utilizing an external power source (BK-Precision 9104), which supplied a direct current density of 0.7 A·cm$^{-2}$ at 5 V for a duration of 15 minutes at 25˚C. Subsequent to deposition, the samples underwent rinsing with acetone and deionized water, followed by air-drying at room temperature. It is worth noting that we have assumed that Zn coating results in a homogeneous layer of constant thickness and no pores. This of course, may not be entirely true and may affect the detection.

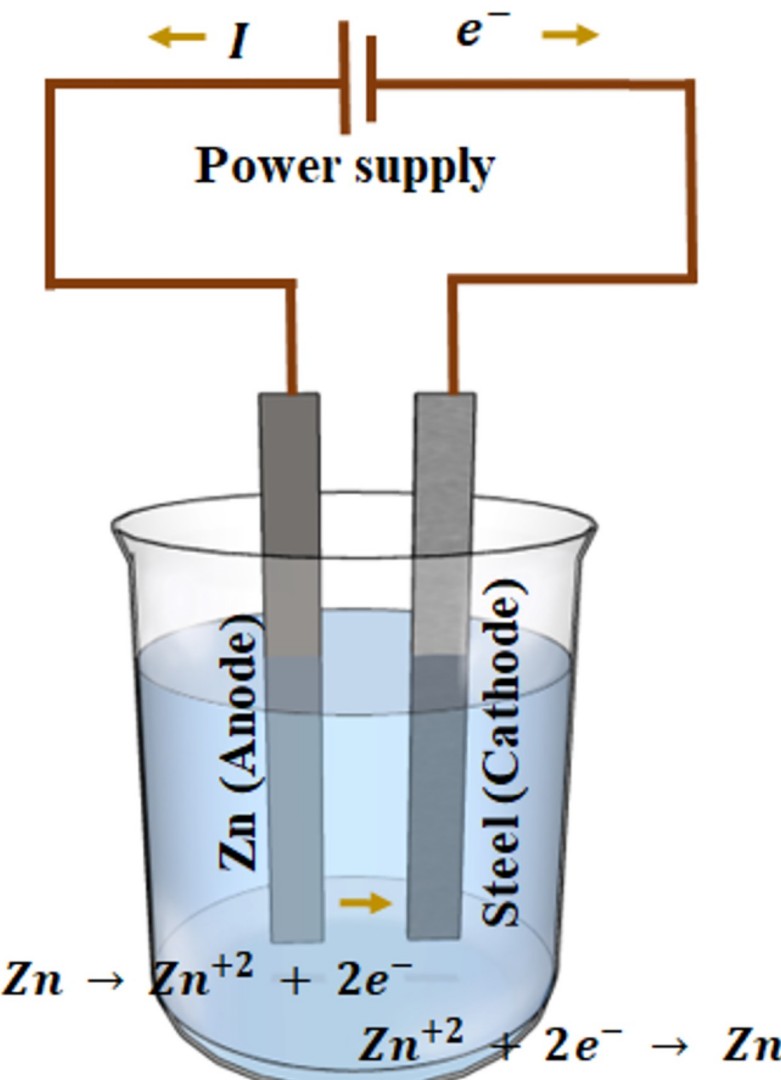

**Fig 3. Diagram of a typical zinc electrodeposition process.** During electroplating, the Zn anode, releases Zn cations into the electrolyte solution due to the flow of electric current. These Zn cations travel through the electrolyte and are attracted to the shim steel, where the Zn cations gain electrons from the external power supply, leading to the deposition of zinc metal onto the shim steel surface. The beaker container holds the electrolyte, which is a 1:1 volume ratio of 15 wt. % HCl acid solution and deionized water.

## The electrochemical cell and the OCP measurements

The metallic sample (AISI 1010 steel shim stock with one side coated with zinc) was attached on the mechanical device's fixed claw, verifying that the steel side was exposed to the electrolyte and the Zn side was insulated and glued to the to the acrylic material (PMMA) from which the components of the device were manufactured. The metallic sample served as the working electrode, a silver/silver chloride reference-electrode (RE) (Sigma-Aldrich) was placed adjacent to the lower claw and a graphite rod was used as the counter electrode. The electric connections, to measure the steel Open Circuit Potential (*OCP*), were mounted as depicted in Fig 1.

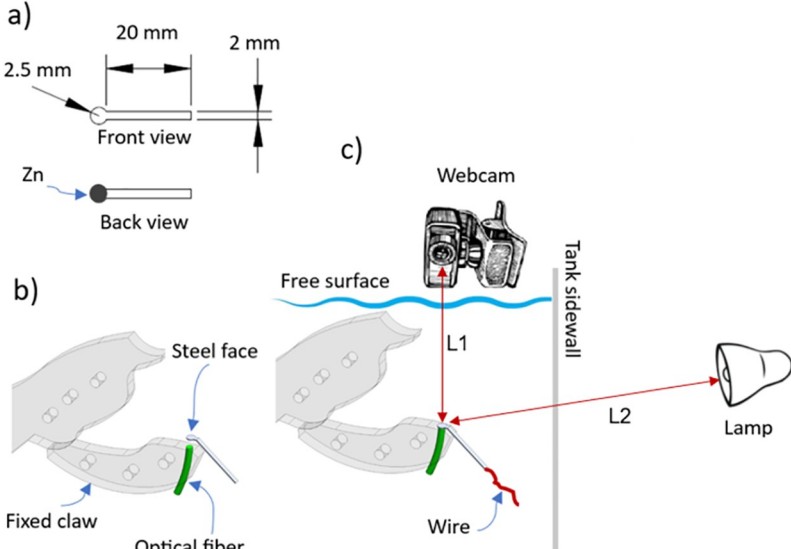

**Fig 4.** a) Geometrical characteristics of the samples and b) their installation in the fixed claw. c) Positioning of the webcam, and of the lamp to visualize during each test the surface of the sample exposed to the flow.

It is important to note that a waterproof-coated copper wire (refer to Fig 4b) was employed to connect the sample to the potentiostat. A rectangular tab on the sample was welded to the wire and this joint was insulated with epoxy resin to prevent any liquid filtrations that might impact the electrochemical measurements.

The exposed area (circular shape, see Fig 4a) of the working electrode was about 0.126 cm$^2$, while the global volume of a 2 wt. % HCl solution was 19 L. Before exposing the working electrode to the corrosive medium, the area was ground using 800 grit silicon carbide paper, cleaned with soapy water, rinsed with distilled water, and degreased with acetone. The *OCP* of the steel surface was measured at a scan rate of 12 dates/min by using a Bio-Logic SP-300 potentiostat.

The *OCP* measurements were performed both in static and dynamic environments (oscillating streams). At the beginning of the test, the sample was immersed in a static solution to obtain the initial OCP for 20 minutes and subsequently, the sample was exposed to oscillatory streams. The duration of the tests was sufficient to allow for the complete perforation of steel (25 $\mu$m thickness) by initial pits. Additionally, extra hours were allocated to study any potentially interesting behaviors.

## Optical configuration and visual tracking of the metallic surface

A description of the optical system is shown in Fig 4. The circular cross-sectional face of the first tip of an optical fiber was placed in direct contact with a green laser beam (532 nm), while the second tip of the fiber was passed through a borehole in the fixed claw. In this way, the circular cross face (through which the laser light is transmitted) of the second tip matched and entered in contact with the back face of the sample where the zinc was electrochemically deposited, see Fig 4a and 4b.

Clear cyanoacrylate glue was employed to uphold and secure contact between the fiber face and the Zn-coated sample face. This method also facilitated a strong fixation of the metal sample to an inner acrylic surface of the fixed claw, preventing fluid penetration around the sample

edges. As a result, the Zn was exposed to laser light while remaining isolated from the external flow.

A Full HD webcam with a resolution of 1080p and a frame rate of 30 fps was used in the tests. The distance between the camera lens and the exposed sample surface was set at $L1 = 12$ cm, see Fig 4c. A USB cable was used to connect the camera to a PC and visualize the sample surface. In addition, the sample surface was illuminated with a 100 W LED lamp. This was placed at about $L2 = 40$ cm from the sample in a position close to one of the tank walls. The webcam was controlled through the specialized SkyStudio Pro software [28] to take pictures every 10 minutes over the course of each experiment. The laser, lamp and electric motor (see Fig 1) were connected to an Arduino UNO board to control the switching on and off of each of these elements (more details on the electrical connections can be found in S2 Appendix).

This allowed the application of duty cycles that included intervals of oscillatory flows (when neither the lamp nor the laser were active), alternating with periods of no flow (static conditions, during which the laser and lamp were alternately switched on and off).

In more detail, each test began with the motor running and thus with the device continuously generating oscillatory flow for three uninterrupted hours. The motor was then turned off for one hour during which no flow was generated, then the motor was turned back on and so on till "$m$" number of flow/no-flow cycles were completed until perforation of the sample was observed. Note that for the test corresponding to Fig 5, $m = 12$ cycles were completed. During each hour of motor rest, the laser was activated for 60 s, followed by deactivation. Subsequently, the lamp was switched on for 540 s, after which it was turned off, initiating a new cycle by reactivating the laser. This was done six times to cover each hour where no flow is generated. The above configurations and settings allowed to follow the time evolution of

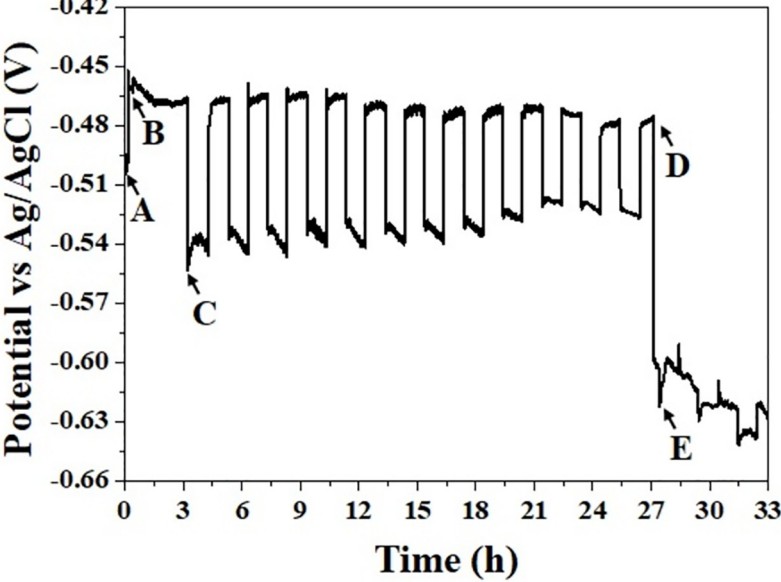

**Fig 5. Open circuit potential of pure steel surface under static and dynamic conditions.** The evolution of *OCP* over the time shows two distinct regions of potential. At flow conditions the potential exhibits a rapid increase from *OCP* initial value (point A) to a higher value (point B). However, when static conditions are imposed, the potential abruptly drops at point C. This cycle of potential increase under flow conditions and potential decrease under static conditions continues until point D. At this point, the potential drops abruptly again (point E), converging towards potentials very close to the *OCP* of the steel/Zn galvanic couple.

corrosion/erosion damage on the samples and to identify over time the laser light spots passing through the specimens.

## Image processing

The images were digitally processed using the Python programming language to determine both the number of pits and the area of the laser spots at different times during the experiment. The digital processing consisted of a series of steps:

i) Converting the photos into monochrome images,

ii) Performing image segmentation using the thresholding method, which involved dividing the image pixels based on an intensity threshold related to image discontinuities,

iii) Applying a filtering process with various masks [29] to define the study area while avoiding edges (a mask was created for each processed image at a specific time due to the displacement induced by the mechanical vibrations of the device), and

iv) Edge detection, which entailed identifying discontinuities in the segmented image (white pixels) and enclosing them with lines related to the laser dots representing pits.

## Determination of the corrosion rate under activation control and no flow conditions

While the primary focus of this research pertains to localized corrosion under oscillatory acidic streams, this section elucidates the methodology for calculating corrosion rates under non-flow conditions, assuming uniform corrosion. This is undertaken to establish a theoretical framework for comparison with the steel dissolution rate in actual operating conditions, providing insights into the potential aggressiveness of the conditions generated with the mechanical device.

Both the polarization resistance $R_p$ and the Tafel extrapolation methods were utilized to determine the corrosion rate of ISI 1010 shim stock steel at room temperature in a 2.0% wt HCl solution. This was done using a three-electrode arrangement and a Bio-logic potentiostat (SP-300). A graphite rod served as counter electrode, while an Ag/AgCl electrode was used as reference electrode. For the anodic and cathodic polarizations in the $R_p$ method, the conditions involved a polarization of $\pm 0.025$ V starting from the potential of corrosion, with a scan rate of 0.001 V/min. The instantaneous corrosion current density, denoted as $i_{corr}$, was determined using Eq 3 [30, 31], where $\beta_a$ and $\beta_c$ represent the anodic and cathodic Tafel slopes (commented below in the Tafel extrapolated method), respectively, while $R_p$ stands for Polarization Resistance in ohms. This resistance is defined as the ratio between the applied voltage and the variation in current $i$ when the metal experiences slight polarization (0.020–0.05 V).

$$i_{corr} = \frac{\beta_a \beta_c}{2.3(\beta_a + \beta_b)R_p} \tag{3}$$

Regarding the Tafel extrapolation method, $i_{corr}$ is graphically obtained by extrapolating the linear segments of both anodic and cathodic reactions up to the corrosion potential on the $E$ vs. $log(i)$ plane. Such linear segments have $\beta_a$ and $\beta_c$ slopes, respectively, which were calculated with a polarization of 0.3 V and 0.6 V from the corrosion potential in both cathodic and anodic directions, under a scan rate of 0.006 V/min.

In order to calculate the corrosion rate (*CR*), Faraday's law of electrolysis was used as expressed in Eq 4 [32]. Here, *M* represents the atomic mass of iron (58.845), *n* denotes the valence change, *F* is Faraday's constant, and $\rho$ is the density of iron (7.874 g/cm$^3$).

$$CR = \frac{i_{corr}M}{nF\rho} \tag{4}$$

At this point, it is important to bear in mind that, while Rp and Tafel extrapolation tests provide swift and effective corrosion rate assessments under static conditions, there exist additional electrochemical techniques such as Electrochemical Impedance Spectroscopy (EIS) that furnish valuable insights into corrosion dynamics. EIS accomplishes this by scrutinizing intricate impedance characteristics across a spectrum of frequencies, promising to unveil nuanced insights into the electrochemical dynamics of such systems. The establishment of an equivalent circuit further refines the interpretative framework, providing a comprehensive understanding of corrosion phenomena, even in dynamic environments and non-aqueous electrolytes [33, 34]. However, while electrochemical methods provide valuable insights into corrosion behavior, the extrapolation of these results to real-world conditions requires careful consideration of the differences between laboratory setups and actual application environments. Many real-world applications involve dynamic conditions, such as cyclic loading, temperature fluctuations, pulsatile flows, etc. These dynamic factors can influence corrosion mechanisms and rates differently than static laboratory conditions. To the best of the authors' knowledge, there is no existing corrosion test conducted under oscillatory streams. Nonetheless, it is important to mention that while intermittent Rotatory Disc Electrode tests could potentially be compared with the current device performance, it is anticipated that distinct results may arise due to the inherent differences in the nature of each device.

## Results and discussion

### Corrosion rate under activation control

In relation to the $R_p$ analysis, the corrosion rate was calculated as 0.6771 $\frac{mm}{year}$, while employing the Tafel extrapolation method resulted in a corrosion rate of 0.7582 $\frac{mm}{year}$. As is well known, each method presented its own set of advantages and disadvantages. For instance, the $R_p$ method offers a trend of the corrosion rate at each moment during an experiment. Theoretically, it is not necessary to polarize the sample beyond ±0.020 V from the corrosion potential, whether in the anodic or cathodic direction. Nevertheless, as indicated by Eq 3, in order to calculate *CR* also the Tafel slopes are necessary, requiring broader polarization values. On the other hand, the Tafel extrapolation method necessitated the extrapolation of the linear portions of both anodic and cathodic reactions up to the corrosion potential to determine $i_{corr}$. Subsequently, the corrosion rate was computed using Faraday's law (Eq 4). The average corrosion rate obtained was 0.7177 $\frac{mm}{year}$, which is lower than the corrosion rate reported by Kousar *et al*. [35] (3.75 ± 0.07 $\frac{mm}{year}$) in 1 M HCl. It is important to note that our experiments were conducted in a 2.0 wt% HCl solution, which is approximately half the concentration used by Kousar [35]. This difference in acid concentration may account for the observed variation in the corrosion rate. In fact, Alzahrani *et al*. [36] reported that the corrosion rate of low carbon steel in HCl media is directly proportional to the acid concentration. However, when compared to results obtained in neutral electrolytes such as 3 wt% NaCl (0.0135 $\frac{mm}{year}$ [37]), the corrosion rate was much higher. Notably, it is essential to highlight that the corrosion rate in acidic solutions is approximately 50 times faster than in neutral conditions. In summary, within such a stagnant acidic solution, the sample, with a thickness of 25 microns, would experience perforation

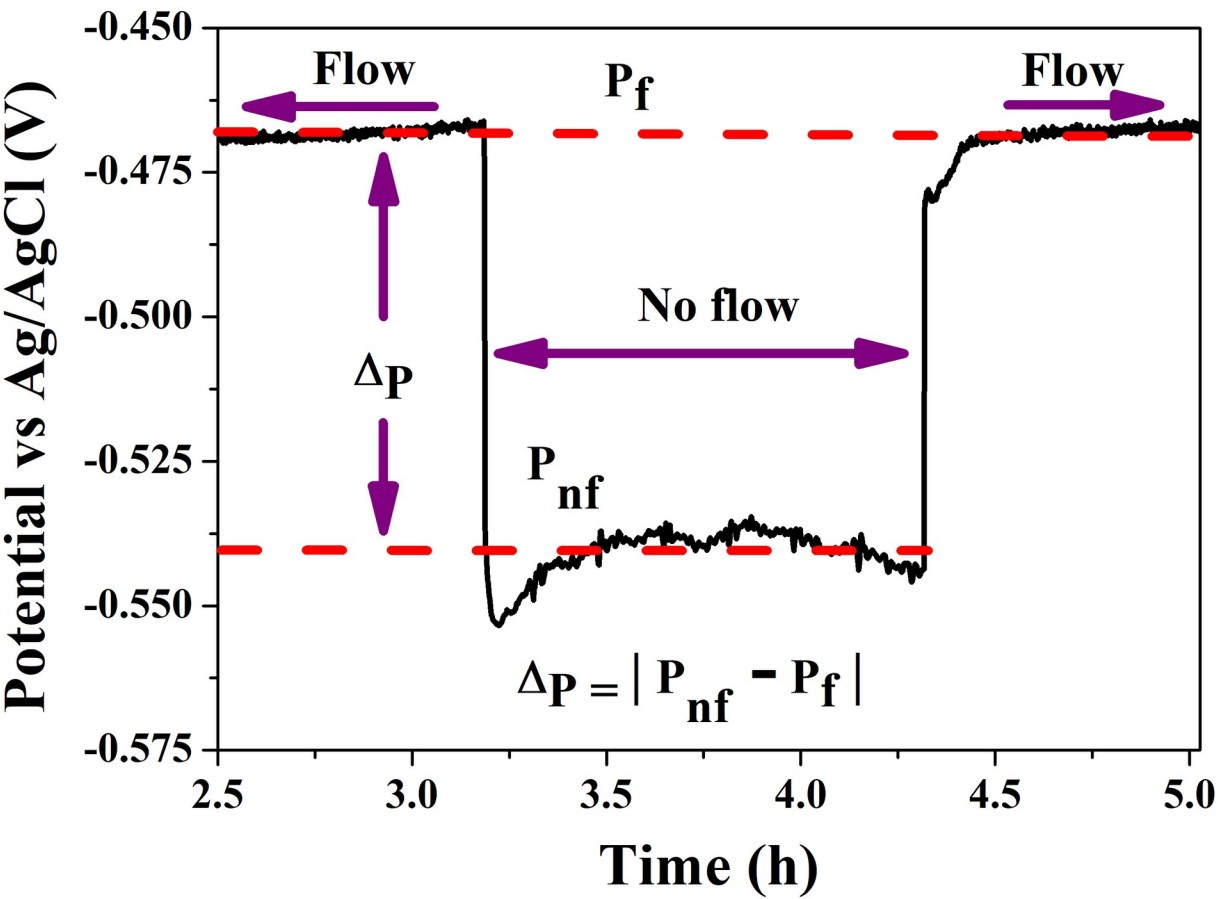

**Fig 6. Details of *OCP* evolution of steel surface in static and dynamic conditions in steel/Zn.** The delta indicates the variation in OCP between different test conditions. It is worth noting that the average potential $P_f$ remains almost constant, whereas the potential $P_{nf}$ exhibits fluctuations over time.

in less than 15 days (precisely 12.5 days), aligning with existing literature. Nonetheless, it is important to bear in mind that these results have been obtained under two very particular considerations: zero-flow conditions and uniform corrosion. It is evident that the expected outcomes under oscillatory streams and in the presence of localized corrosion will be much more aggressive. However, having a benchmark is always crucial for conducting a proper evaluation of such results.

### Electrochemical behavior under oscillatory acidic streams

Fig 5 illustrates the evolution of the open circuit potential (*OCP*) of the iron surface concerning immersion time under flow and no-flow conditions. In both scenarios, distinct regions emerge characterized by two different potentials as shown in Fig 6. During the period subjected to flow conditions the potential exhibits a relatively higher value ($P_f$), while during the period subjected to static conditions (no-flow) the potential displays a comparatively lower value ($P_{nf}$).

As indicated in Fig 5, in the initial moments of the *OCP* measurement, the potential remains consistent around -0.499 V (point A). However, as the mechanical device is activated

and the flow interacts with the steel surface of the sample, the potential goes to -0.4518 V (point B). This shift could be attributed to a reduction in the resistance of the cathodic reaction, stemmed from an increased dissolved oxidizing agent within the electrolyte surrounding the steel sample [38]. Over time, such potential experiences a temporary recovery, eventually stabilizing at an average potential of -0.4683 V during the initial three-hour period.

In contrast, under static conditions, the potential suffers an abrupt drop to -0.5530 V (point C). This alternating behavior persisted until point D, where the potential drops suddenly below -0.6222 V (point E) due to the appearance and growth of pits that penetrated the 25-micron thickness of the steel. In fact, the occurrence of localized corrosion in shim steel indicates that the electrolyte has reached and dissolved a certain amount of Zn content. During the initial localized dissolutions, only a small amount of Zn is reached by the electrolyte (Fig 7 at 26 hours). Therefore, the *OCP* does not decrease. However, as the steel dissolution increases (around an hour), more electrolyte reaches the Zn, causing the *OCP* to decrease to a value between the potential of the steel and Zn. Consequently, the galvanic couple is consolidated at a potential around -0.6222 V [39]. It is essential to note that previously, a steel shim (pre-coated with Zn) underwent perforation using a tiny pin, exposing only a small area of Zn. The resulting open circuit potential was -0.565 V, progressively decreasing to -1.0 V as the Zn area expanded with an increasing number of perforations (once the Zn is exhausted the potential slowly shifts towards less negative values).

This highlights the accuracy of the electrochemical design in detecting with acceptable precision the time at which the steel undergoes complete perforation due to localized corrosion phenomena. In other words, these results affirm the ability of the electrochemical sensor (steel-Zn system) to accurately determine the average rate of pitting progression, which is corroborated by the optical observations discussed in the following section.

As is well known, the behavior of the *OCP* provides information about the thermodynamic stability of the sample surface in corrosive media. Therefore, the trend and the high-frequency spikes in the potential signal represent changes in the oxidation tendency of the surface due to the instability of thousands of chemical reactions at the interface of steel and electrolyte. In reference to Fig 6 (zoom of Fig 5), it can be observed that the potential $P_f$ remains nearly unchanged on average during flow, in contrast to the $P_{nf}$ potential. According to Gou *et al.* [38], the *OCP* remains constant over time due to two situations: either the anodic and cathodic reactions are unaffected by the conditions of *OCP* measurement, or both reactions influence each other, resulting in a constant *OCP*. Rapid flow on the sample surface enhances the rate of oxidizing agent mass transfer to the electrode surface and reduces the thickness of the diffusion boundary layer [40]. As a result, flow increases the rate of reduction reaction on the surface, as well as enhances the anodic reaction leading, under some conditions, to the growth of a passive film. Consequently, in theory, only insignificant changes in potential can be observed [41]. This suggests that the electrolyte flow should not alter the *OCP* obtained under static conditions. Despite the present findings indicating a clear difference ($\Delta_p$) between both $P_f$ and $P_{nf}$, such a difference significantly decreases over time. Both $P_f$ and $P_{nf}$ progressively approach the initial open circuit potential (-0.499 V) as the steel shim undergoes significant dissolution.

Additionally, high-frequency spikes are present and visually similar in both static and dynamic conditions of *OCP* measurements. However, the potential data of the no-flow signal has a higher standard deviation (0.00281) compared to the signal under flow (0.00186). It has been reported that frequent and continuous spikes in the potential signal can be attributed to the uniform corrosion process occurring on the surface of the carbon steel electrodes in acidic media [42]. This indicates a highly unstable electrochemical process on the steel surface when the standard deviation value is high [43]. In a certain sense, this explains why the sample under no-flow conditions exhibits a higher tendency to uniform corrosion (early corrosion

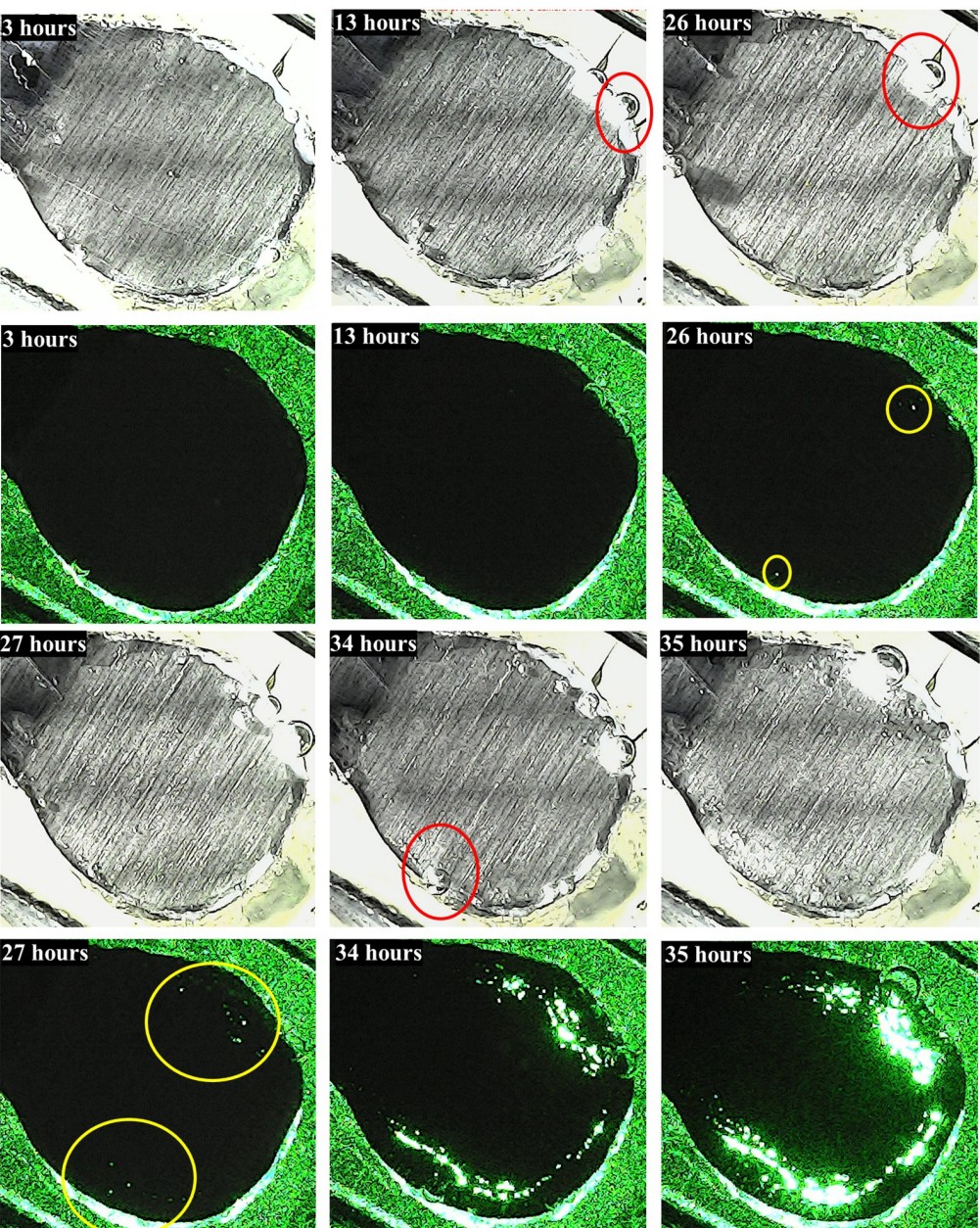

**Fig 7. White light images and corresponding images using only laser illumination of a sample under the effect of oscillatory acidic flows.** The white light images are taken to provide an overview of the sample and to identify areas where hydrogen gas bubbles are being generated. Red circles around the bubbles help to distinguish them from the rest of the sample.

stage). However, as the experimental evolved, a high susceptibility to localized corrosion was present, as shown in Figs 7 and 8. In these instances, the acid flow on the steel surface plays an important role in this type of corrosion, due to defects inherent to the steel shim (such phases in the metal matrix and surface scratches [44]) and defects generated by the experimental set-up. These factors likely alter the shear stresses and contribute to the occurrence of localized corrosion.

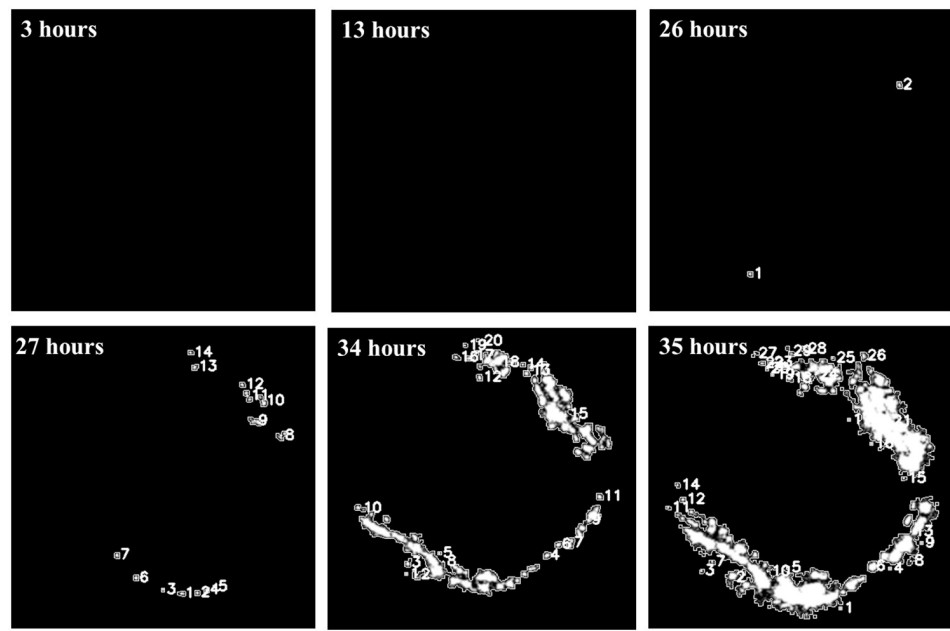

**Fig 8. Sequence of digital images showing the corrosion process over time caused by pulsating acidic streams.** At 26 hours, the first pits appear (two pits); as the time of the experiment passes, a considerable increase in new pits is identified on the surface.

## Corrosion behavior by image analysis

Top views of the metallic samples under the electrolyte are presented in Fig 7 with and without white illumination in order to better appreciate the green laser light located beneath the metallic sample. As anticipated, during the initial hours of experimentation, it was not possible to discern any laser spot, indicating that the steel sample remained free from perforations generated by pits. Nonetheless, as the exposure time in the electrolyte continued, small laser spots were registered after 26 hours. What is interesting to note is that the first pits were observed in areas close to the edges of the sample. As previously mentioned, cyanoacrylate was successfully used to fix the sample and prevent leakage of the electrolyte. It is worth stressing that no filtrations of electrolyte were detected at the metal/coating interface of the work electrodes. Otherwise, the electrochemical potential would change abruptly before the generation of pits could occur. However, we recognize the possibility of having left unevenly sized resin edges around the samples, which may affect electrolyte flow and shear stresses on the exposed metal. The lack of symmetry in these imperfections suggests non-uniform corrosion at the edges and in the area exposed to flow. Sample geometry and size may also contribute to corrosion near the boundaries. Future research will certainly need to investigate the flow velocity near the sample, using techniques such as PIV, and complement with CFD simulations to better understand the impact of geometry and edges as well as the corrosion mechanisms that give rise to pitting in these zones. Beyond 26 hours, the number of pits increased, becoming more discernible from the surrounding non-corroded areas. On the other hand, images captured by using white illumination revealed (from the 13 hours) the generation of $H_2$ (red circles in Fig 7) resulted from the reduction of the $H^+$ ions present in the acidic electrolyte. Such a cathodic reaction indicates that its counterpart (the steel dissolution) has already started. In short, previous remarks prove the complementarity of both kinds of images. On the one hand, images with

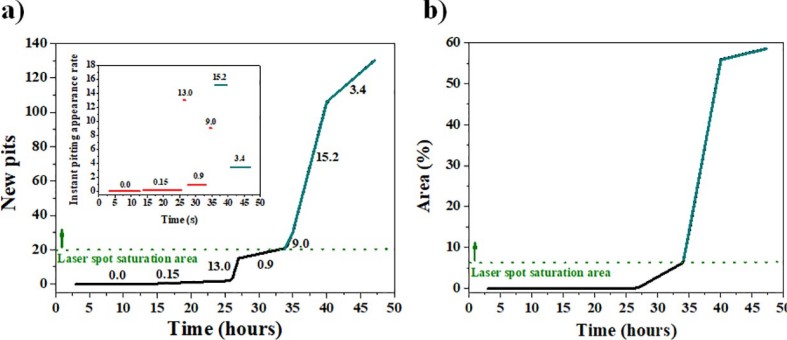

**Fig 9. a)** Number of new pits (the inset illustrates the instant pitting appearance rate in pits per hour) and, **b)** area of active points from digital analysis of Zn coated steel shim.

white illumination are not able to reveal the first pits but the first hydrogen bubbles, whereas dark images (with the green laser activated) certainly provide quantitative information of the generation of consolidated pits but nothing about the cathodic reaction.

To achieve satisfactory detection results for hidden pitting, the images underwent digital processing, as depicted in Fig 8. Thresholding was used as the segmentation process by assigning a binary value to each pixel on the studied area. As is well known, such a process is based on pixel intensities and a limit value determined by the intensity of light (passing through the sample in this work). This process aimed to create a sharp contrast between foreground objects (active points) and the background. It is worth noting that the white area represents the laser spot allowed by pitting corrosion, stemming from a rapid localized corrosion mechanism that led to substantial material loss (both in steel and Zn) in very confined regions. Such a non-uniform corrosion, manifested as pits, arises from the non-homogeneous nature of the electrode surface and the accumulation of stress induced by repeated mechanical load cycling. Upon comparing both Fig (Figs 7 and 8), it becomes evident that counting the number of pits through visualizing the green spots becomes unfeasible even at 34 hours due to light saturation issues. Nevertheless, employing digital image processing allows for pit counting even at 35 hours, albeit with a certain degree of uncertainty.

Fig 9 illustrates the progression of both the number of pits and the area of active zones over time. Starting from 34 hours, the results obtained from the quantification analysis exhibit a high degree of uncertainty due to the extensive overlap of different contributions from active points (active zones). This is attributed to the saturation of the laser spot, wherein the brightness exceeds the camera sensor capacity. Consequently, the region of laser spot saturation was delineated as an area where associating the number of pits with the laser spot becomes unfeasible. Below such a saturation region, the quantification was much clearer.

The pitting appearance rate represented by the 1st derivative of the number of pitting curve are presented in the inset of Fig 9a. The value of each horizontal red line in the inset corresponds to the slope of each black line in the main image. Although there is no a clear correlation between the occurrence rate of pits and the duration of exposure, it is possible to establish an average rate of 0.61 pits per hour before reaching the saturation threshold. This number increases significantly when accounting for uncertainty and considering 45 hours of exposure (2.73 pits per hour). In light of the fact that the laser intensity remained constant throughout the entire experimental duration, the spot area in pixels (Fig 9b) stands as a parameter with reduced measurement uncertainty, given that the saturation area is sufficiently low

(approximately 6%). As a matter of fact, it has been documented [45] that the method of laser detection through image saturation (employed in this study) relies on a predetermined threshold that is fine-tuned until the method yields a binary image with fewer than 10% white pixels.

## Conclusion

The most important findings of this research are summarized below:

- The significant disparity (26 hours vs. 12.5 days, respectively) found when comparing the experimental perforation time under acidic flow-induced corrosion with the time under no-flow conditions (assuming homogeneous corrosion) vividly illustrates the aggressiveness of the conditions induced by the mechanical device.

- The bio-inspired claw device, coupled with an innovative electrochemical cell design, enables Open Circuit Potential (OCP) tracking during acidic streams. This approach offers quantitative data on perforation points and pit growth rate approximations.

- The formed galvanic couple between the steel surface and zinc coating exhibits a distinctive potential, providing a clear differentiation from the corrosion potential of the steel, even with initial pits.

- Both electrochemical tracking and optical methods demonstrate reliability, with a one-hour maximum discrepancy in occurrence times. This reliability is crucial in understanding the complex and stochastic nature of flow-induced corrosion and erosion processes.

- The illumination method not only verifies perforations but also precisely pinpoints pit locations on the sample.

- The amalgamation of electrochemical tracking, optical setup, and image processing provides a powerful tool for corrosion detection and assessment.

In order to improve on and refine the methods presented here, it would be interesting to investigate the following important subjects in the future: different approaches to sample fixing and sealing; alternative metal deposition techniques and galvanic couples to guarantee constant coating thicknesses and homogeneity; application of microscopes for early pitting detection; and a thorough examination of the light spots to correlate their properties with the dimensions and configuration of the material perforations. In conclusion, this research opens exciting possibilities for effective corrosion monitoring and prevention across various industrial and research sectors, with potential applications in turbulent flows and biomedical fields, just to name a few.

## Supporting information

**S1 Appendix. Calculation of flow rate.**
(PDF)

**S2 Appendix. Circuit to control the duty cycles of the motor-laser-lamp system.**
(PDF)

**S1 Fig.** a) Mouthpiece area vs. time. b) Different opening positions of the mobile claw to show how the surface *A* (represented by the blue surface) varies as a function of the angular position and therefore of time. In position (1) the claw is in its initial condition with the smallest mouthpiece area, in position (2) the maximum opening is shown and consequently also *A* is maximum. c) Volume of the socket formed by the internal side walls of the two support plates (6), as shown in Fig 1, the internal side walls of both the fixed and mobile claws and of course

the area *A* of the mouthpiece. d) Two opening positions of the mobile claw to indicate the area *A*1 (shown in purple color), as well as the internal volume of the socket, the latter corresponding to the intermediate position (3). Clearly, the volume of the socket changes with the theta angle and also with time. The symbols on the $A(t)$ and $V(t)$ curves correspond to data obtained from area and volume measurements at different angular positions, respectively; the solid lines in both plots represent interpolations using cubic splines to smooth their appearance.
(TIF)

**S2 Fig. Connections and components of the circuit used to control the switching on and off of the devices.** Relays to control the a) motor, b) laser, and c) lamp. d) Arduino UNO microcontroller, e) 532 nm (50 mW) laser, f) DC-DC boost converter to power the laser, g) 3mm diameter PMMA fiber optic cable.
(TIF)

## Author Contributions

**Conceptualization:** Francisco A. Godínez, Marvin Montoya-Rangel, Rodrigo Montoya.

**Data curation:** Francisco A. Godínez, Marvin Montoya-Rangel, Rodrigo Montoya.

**Formal analysis:** Francisco A. Godínez, Marvin Montoya-Rangel, Rodrigo Montoya.

**Funding acquisition:** Francisco A. Godínez, Rodrigo Montoya.

**Investigation:** Francisco A. Godínez, Marvin Montoya-Rangel, Rodrigo Montoya.

**Methodology:** Francisco A. Godínez, Marvin Montoya-Rangel, Rodrigo Montoya.

**Project administration:** Francisco A. Godínez, Rodrigo Montoya.

**Resources:** Francisco A. Godínez, Marvin Montoya-Rangel, Rodrigo Montoya.

**Software:** Francisco A. Godínez, Marvin Montoya-Rangel, Rodrigo Montoya.

**Supervision:** Francisco A. Godínez, Rodrigo Montoya.

**Validation:** Francisco A. Godínez, Marvin Montoya-Rangel, Rodrigo Montoya.

**Visualization:** Francisco A. Godínez, Rodrigo Montoya.

**Writing – original draft:** Francisco A. Godínez, Marvin Montoya-Rangel, Rodrigo Montoya.

**Writing – review & editing:** Francisco A. Godínez, Rodrigo Montoya.

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
