## [Decision Letter · Decision Letter 0]

21 Nov 2023

PONE-D-23-36475Accelerated corrosion of low carbon steel by oscillatory acidic streams generated with a bio-inspired claw devicePLOS ONE

Dear Dr. Montoya,

Thank you for submitting your manuscript to PLOS ONE. After careful consideration, we feel that it has merit but does not fully meet PLOS ONE’s publication criteria as it currently stands. Therefore, we invite you to submit a revised version of the manuscript that addresses the points raised during the review process.

ACADEMIC EDITOR: Please, the Reviewers have suggested to a REVISION. Although, it seems that the manuscript deserves its publication, before a rebuttal letter detailing/commenting all suggestions should be provided.

We look forward to receiving your revised manuscript.

Kind regards,

Wislei Riuper Osório

Academic Editor

PLOS ONE

Journal Requirements:

3. Please expand the acronym “UNAM-DGAPA” (as indicated in your financial disclosure) so that it states the name of your funders in full.

4. Thank you for stating the following financial disclosure: "UNAM-DGAPA-PAPIIT IG100623 program has financed this work. MMR

thanks UNAM-DGAPA for the postdoctoral fellowship throughout the program POSDOC."

5. Thank you for stating the following in the Acknowledgments Section of your manuscript: "UNAM-DGAPA-PAPIIT IG100623 program has financed this work. Montoya-Rangel thanks UNAM-DGAPA for the postdoctoral fellowship throughout the program POSDOC"

Please remove any funding-related text from the manuscript and let us know how you would like to update your Funding Statement. Currently, your Funding Statement reads as follows: "UNAM-DGAPA-PAPIIT IG100623 program has financed this work. MMR

thanks UNAM-DGAPA for the postdoctoral fellowship throughout the program POSDOC."

7. Please upload a copy of Supporting Information Figure/Table/etc. Supporting Information S1 Appendix and S2 Appendix which you refer to in your text on page 12.

Reviewers' comments:

Reviewer's Responses to Questions

**Comments to the Author**

1. Is the manuscript technically sound, and do the data support the conclusions?

Reviewer #1: Yes

Reviewer #2: Partly

Reviewer #3: Yes

Reviewer #4: Yes

2. Has the statistical analysis been performed appropriately and rigorously? 

Reviewer #1: N/A

Reviewer #2: No

Reviewer #3: Yes

Reviewer #4: Yes

3. Have the authors made all data underlying the findings in their manuscript fully available?

Reviewer #1: Yes

Reviewer #2: No

Reviewer #3: No

Reviewer #4: Yes

4. Is the manuscript presented in an intelligible fashion and written in standard English?

Reviewer #1: Yes

Reviewer #2: Yes

Reviewer #3: Yes

Reviewer #4: Yes

5. Review Comments to the Author

Reviewer #1: Dear authors, the manuscript is overall well written and the results are well discussed. However, I have some questions and suggestions before it is published in PLOS ONE. Below I present my points:

- The manuscript is appropriate for the subject matter. However, some sections could be simplified for better readability without sacrificing technical accuracy. Consider rephrasing complex sentences for clarity, particularly in the methodology section. A thorough proofreading for punctuation (such as comma usage) and sentence structure is recommended to enhance readability.

- The use of a webcam and lamp for visualization is mentioned, but additional information on the exact setup, calibration, and limitations of this approach would strengthen this section.

- In the results section, you've presented corrosion rates obtained through different methods (Rp analysis and Tafel extrapolation). How do these rates compare with those observed in real-world conditions, and what might be the reasons for any discrepancies? Can you provide more insights into how the performance of this device compares with existing corrosion testing methods? Additionally, could you elaborate on the implications of these findings for the practical application of your device?

- Establishing a theoretical framework for comparison is a good approach, but the manuscript might benefit from a more detailed explanation of this framework, including any assumptions or limitations.

- The paper would be better with a stronger conclusion section, succinctly summarizing the findings, following discussions of implications in the broader context of corrosion research and practical applications.

Reviewer #2: INTEREST:

Dear Editor,

I am writing to express my interest in the manuscript titled "Accelerated corrosion of low carbon steel by oscillatory acidic streams generated with a bio-inspired claw device" by Francisco A. Godínez and colleagues. The research discusses a mechanical device inspired by the pistol shrimp's claw, designed to create oscillating flows for studying corrosion rates in carbon steel. It details an innovative method using thin steel samples coated with zinc for electrochemical detection of corrosion, complemented by laser illumination and imaging to locate and count corrosion pits. I believe that the PLOS ONE Journal would be an appropriate venue for this work. However, to meet the exacting standards of this journal, there are key areas in which the manuscript could benefit from further refinement, especially in its presentation and analysis of results. I advocate for a thorough revision, taking into account the detailed feedback provided below.

COMMENTS:

Please consider the following specific recommendations for revision:

a) Why did the authors choose to use an electrolyte composed of a 1:1 volume ratio of 15 wt% HCl acid solution and deionized water for Zn electrodeposition on 1010 steel? Why didn't they use other baths that have much higher current efficiency?

b) The corrosion science literature shows that for a material to undergo localized pitting corrosion, it must be in a passive state. And, as I read the manuscript, I became a little confused, as the authors at some points in the manuscript talk about localized corrosion, metastable pits, repassivation, and at others about generalized corrosion. Here are some questions: (i) Is 1010 carbon steel in the passive state in a 2 wt. % HCl solution???; (ii) For such an explanation, it would be extremely important for the authors to present both the potentiodynamic polarization curves and the linear polarization curves (used to determine the Rp values, in the 2 wt. % HCl solution. Given the answer presented by the authors, perhaps it would be important to review the following statement contained in the manuscript “...It has been reported that spikes in the potential signal can be related to the occurrence of metastable pits and rapid repassivation in an electrochemical process [38] [39] [40], indicating a highly unstable electrochemical process on the steel surface when the standard deviation value is high [41]. This, in a way, elucidates why the sample under flow conditions exhibits reduced susceptibility to repassivation and a greater propensity for the pitting process...".

c) Before the exposure to the corrosive medium, how were the samples prepared? Were they polished, cleaned, or treated in any way? Surface conditions can significantly influence degradation behaviors.

d) When looking at Figure 7 of the manuscript, I get the impression that there are crevices forming at the metal/coating interface of work electrodes. Was there any special care taken to avoid this type of problem when isolating the area to be exposed to the corrosive environment? It would be important for the authors to make some mention of this and say what type of error this could lead to in the measurements.

e) When discussing Figure 5, the authors make the following comment: “...In contrast, under static conditions, the potential suffers an abrupt drop to -0.5530 V (point C). This alternating behavior persisted until point D, where the potential drops abruptly below -0.6222 V (point E) due to the appearance and growth of pits that penetrated the 25-micron thickness of the steel, exposing tiny surfaces of Zn. Subsequently, the signal converges towards potentials very close to OCP of the steel/Zn galvanic couple…”. To support this statement, why don't the authors perform potentiodynamic polarization measurements for the Zn coating and then superimpose these measurements on the steel polarization measurements? This way, the authors could empirically prove the value of the Zn/steel galvanic couple.

g) In my opinion, the conclusion must be more objective, reflecting the outcome of the work. Please be more concise and critical, highlighting the key findings. Additionally, it would be beneficial to offer suggestions for subsequent research and potential advancements in this domain.

Reviewer #3: Dear author,

The paper is interesting. In my frank opinion, it deserves to be published after minor revisions:

Minor revision of English is necessary.

The images are not in good resolution.

Viscosity can be changed with pressure, temperature and concentration. In formula 2, wouldn't it be necessary to take these variables into account within viscosity?

Why exactly use the green laser? Wouldn't it be more interesting to use a laser in the UV region? The UV laser simulates sunlight.

“The images were digitally processed using Python Programming Language to determine”. It would be interesting if you provided, as an attachment, the programming used.

“the polarization resistance Rp and the Tafel extrapolation methods were utilized to determine the corrosion rate”. How the externally induced mechanical oscillation affects the charge transfer of the system: In the Tafel extrapolation, the frequency of the electric current through the material is taken into account.

Example: formula 6:

https://www.mdpi.com/2227-9717/11/3/721

The study you did is very interesting and important. To emphasize the importance of this study, I suggest mentioning, for example in the introduction, other techniques for studying corrosion (currently used in static media, such as Electrochemical Impedance Spectroscopy), fuels such as electrolytes and different material manufacturing routes. Below are examples of articles that illustrate these points.

https://journals.sagepub.com/doi/abs/10.1177/00219983211029352

https://meridian.allenpress.com/corrosion/article-abstract/76/12/1109/447822/Silicon-Content-Affecting-Corrosion-Behavior-of

Reviewer #4: Dear authors, below are some comments and suggestions that I consider relevant to the work and doubts that may not have been clarified in the text:

1. It is known that the central objective is not to be the result of the corrosion test but there is a focus on this area as well. It is believed to be interesting to think about applying an equivalent circuit to obtain relevant information regarding corrosion parameters. Below is an indication of reading the articles: GARCIA, A. et al (2009) Electrochemical corrosion response of a low carbon heat treated steel in a NaCl solution. and GARCIA, A. et al. (2010) Electrochemical corrosion behaviour of a Ti-IF steel and a SAE 1020 steel in a 0.5M NaCl solution., where corrosion in steel is discussed.

2. I believe the explanation for the galvanic couple formation after the first pit appearance occurs is vague. Something expected did not happen.

3. The analysis of the formation of pits was never discussed regarding their appearance on the edges of the samples. Is there any influence on the sample preparation or its dimensions?

6. PLOS authors have the option to publish the peer review history of their article (what does this mean?). If published, this will include your full peer review and any attached files.

Reviewer #1: No

Reviewer #2: **Yes: **Carlos Alberto Della Rovere

Reviewer #3: **Yes: **Prof. Dr. Yuri Alexandre Meyer

Reviewer #4: No

---

## [Author Response · Author response to Decision Letter 0]

20 Dec 2023

Reviewer #1: Dear authors, the manuscript is overall well written and the results are well discussed. However, I have some questions and suggestions before it is published in PLOS ONE. Below I present my points:

- The manuscript is appropriate for the subject matter. However, some sections could be simplified for better readability without sacrificing technical accuracy. Consider rephrasing complex sentences for clarity, particularly in the methodology section. A thorough proofreading for punctuation (such as comma usage) and sentence structure is recommended to enhance readability.

Thanks a lot for the comments, the authors appreciate the time and effort put in reviewing our work.

Some paragraphs in the methodology section were rephrased to make them more precise. They are listed below for better location in the revised manuscript.

A) It is important to note that a waterproof-coated copper wire (refer to Figure 4b) was employed to connect the sample to the potentiostat. A rectangular tab on the sample was welded to the wire and this joint was insulated with epoxy resin to prevent any liquid filtrations that might impact the electrochemical measurements.

B) Clear cyanoacrylate glue was employed to uphold and secure contact between the fiber face and the Zn-coated sample face. This method also facilitated a strong fixation of the metal sample to an inner acrylic surface of the fixed claw, preventing fluid penetration around the sample edges. As a result, the Zn was exposed to laser light while remaining isolated from the external flow.

C) This allowed the application of duty cycles that included intervals of oscillatory flows (when neither the lamp nor the laser were active), alternating with periods of no flow (static conditions, during which the laser and lamp were alternately switched on and off).

D) During each hour of motor rest, the laser was activated for 60 s, followed by deactivation. Subsequently, the lamp was switched on for 540 s, after which it was turned off, initiating a new cycle by reactivating the laser.

E) The deposition area was standardized at 1 cm2, by masking the remaining area of one side of the electrode and the entirety of the opposite side with 3MTM electroplating tape, with a 1.5 cm separation between the zinc electrode and the steel shim.

D) The images were digitally processed using the Python programming language to determine both the number of pits and the area of the laser spots at different times during the experiment. The digital processing consisted of a series of steps:

i) Converting the photos into monochrome images,

ii) Performing image segmentation using the thresholding method, which involved dividing the image pixels based on an intensity threshold related to image discontinuities,

iii) Applying a filtering process with various masks [29] to define the study area while avoiding edges (a mask was created for each processed image at a specific time due to the displacement induced by the mechanical vibrations of the device), and

iv) Edge detection, which entailed identifying discontinuities in the segmented image (white pixels) and enclosing them with lines related to the laser dots representing pits.

- The use of a webcam and lamp for visualization is mentioned, but additional information on the exact setup, calibration, and limitations of this approach would strengthen this section.

We welcome this suggestion. In response, we have added the following paragraphs to the revised manuscript:

“A Full HD webcam with a resolution of 1080p and a frame rate of 30 fps was used in the tests. The distance between the camera lens and the exposed sample surface was set at L1=12 cm. A USB cable was used to connect the camera to a PC and visualize the sample surface. In addition, the sample surface was illuminated with a 100 W LED lamp. This was placed at about L2= 40 cm from the sample in a position close to one of the tank walls.”

Note that figure 4 has been slightly modified to indicate with arrows the distances L1 and L2.

- In the results section, you've presented corrosion rates obtained through different methods (Rp analysis and Tafel extrapolation). How do these rates compare with those observed in real-world conditions, and what might be the reasons for any discrepancies?

Corrosion rates obtained through different methods, such as Rp analysis and Tafel extrapolation, may not always directly correlate with the corrosion rates observed in real-world conditions. There are several reasons for discrepancies between laboratory measurements and real-world scenarios:

Laboratory experiments often use controlled and simplified conditions that may not fully represent the complexity of real-world environments. Real-world conditions can involve various factors like temperature variations, impurities, microbial activity, and flow dynamics, which may influence corrosion differently than controlled laboratory conditions.

Surface preparation in laboratory studies may differ from real-world surfaces. The presence of contaminants, oxides, or passive layers on real-world surfaces can impact the corrosion resistance differently than clean and polished surfaces used in controlled experiments.

Corrosion rates obtained in laboratory experiments are often measured over relatively short periods. In real-world conditions, corrosion can occur over extended periods, and long-term exposure may lead to different corrosion mechanisms or rates.

Many real-world applications involve dynamic conditions, such as cyclic loading, temperature fluctuations, and variations in humidity. These dynamic factors can influence corrosion mechanisms and rates differently than static laboratory conditions.

In summary, while laboratory methods like Rp analysis and Tafel extrapolation provide valuable insights into corrosion behavior, the extrapolation of these results to real-world conditions requires careful consideration of the differences between laboratory setups and actual application environments. Conducting field tests or using advanced modeling techniques that account for a broader range of environmental variables can help improve the accuracy of corrosion rate predictions in practical applications.

 Can you provide more insights into how the performance of this device compares with existing corrosion testing methods? 

To the best of the authors' knowledge, there is no existing corrosion test conducted under oscillatory streams. However, it is noteworthy that the Rotating Disc Electrode (RDE) serves as a specialized electrochemical tool designed for the study of electrochemical reactions, particularly those occurring at the interface between an electrode and a solution under continuous flow conditions. This technique is highly valuable for investigating reaction kinetics, mass transport, and electrode processes. The rotating disc enhances mass transport, thereby mitigating concentration polarization effects, making it particularly instrumental in comprehending processes involving mass transport limitations or complex reaction mechanisms. So, while intermittent RDE tests could potentially be compared with the current device performance, it is anticipated that distinct results may arise due to the inherent differences in the nature of each device.

Additionally, could you elaborate on the implications of these findings for the practical application of your device?

On one hand, the combination of two original and independent methodologies—electrochemical and optical—has been developed to detect the exact moment when a metallic sample (of specific thickness) is fully perforated due to corrosion. The fact that both methodologies have recorded very similar results provides corrosion specialists with a powerful tool applicable in diverse settings for estimating the actual expected life of specific systems, such as pipelines.

On the other hand, as outlined in the introductory section, acid solutions find common usage in various industrial processes, including pickling, cleaning, descaling, and oil-well acidizing. Therefore, the practical application of both methodologies presented in this research is perfectly suited for such areas.

Concerning the mechanical device, as demonstrated previously [10, 18-20], by varying its closure rate it could even induce cavitation, offering a robust tool to generate different types of flows that can replicate various phenomena present in numerous industrial applications.

In summary, the aforementioned reasons pave the way for the development of corrosion rate sensors that register galvanic interactions between two metals (alongside optical laser detection). This introduces an innovative technology capable of efficiently detecting the impact of electrolyte flows under diverse regimes.

- Establishing a theoretical framework for comparison is a good approach, but the manuscript might benefit from a more detailed explanation of this framework, including any assumptions or limitations.

Thank you for this recommendation. The assumptions and limitations that in our opinion are the most important for this work are listed below:

Assumptions

1) Zn deposition gives rise to a homogeneous layer of constant thickness and no pores.

2) The glue used to firmly fix the sample also prevents fluid leakage (through the edges of the sample) underneath the sample into the Zn coating.

3) The glue used to fix the sample does not leave edges on its periphery that could affect the flow characteristics in this area and therefore the corrosive effects.

Limitations

1) The optical technique does not give an exact idea of the size of the pits, since what is detected are spots of light that are not necessarily the same size as the pits.

2) The developed technique does not allow detecting pitting in early stages of formation, only when the material has been perforated. 

3) When the motor is operated at speeds above 250 rpm, mechanical vibrations are induced throughout the device that can hinder both electrochemical and optical sensing.

Please note that the assumptions and limitations were paraphrased, and they were not added as a bullet list but were interspersed throughout the text as best suited.

- The paper would be better with a stronger conclusion section, succinctly summarizing the findings, following discussions of implications in the broader context of corrosion research and practical applications.

We agree with your assessment. In the new version of the manuscript we shortened the conclusions to make them more concise and clear. In order to enhance the suggested methods, we have highlighted a few topics that we think should be further investigated. Possible applications were also briefly mentioned.

Reviewer #2: 

INTEREST:

Dear Editor,

I am writing to express my interest in the manuscript titled "Accelerated corrosion of low carbon steel by oscillatory acidic streams generated with a bio-inspired claw device" by Francisco A. Godínez and colleagues. The research discusses a mechanical device inspired by the pistol shrimp's claw, designed to create oscillating flows for studying corrosion rates in carbon steel. It details an innovative method using thin steel samples coated with zinc for electrochemical detection of corrosion, complemented by laser illumination and imaging to locate and count corrosion pits. I believe that the PLOS ONE Journal would be an appropriate venue for this work. However, to meet the exacting standards of this journal, there are key areas in which the manuscript could benefit from further refinement, specially in its presentation and analysis of results. I advocate for a thorough revision, taking into account the detailed feedback provided below.

First of all thanks a lot for the comments, the authors appreciate the time and effort put in reviewing our work.

COMMENTS:

Please consider the following specific recommendations for revision:

a) Why did the authors choose to use an electrolyte composed of a 1:1 volume ratio of 15 wt% HCl acid solution and deionized water for Zn electrodeposition on 1010 steel? Why didn't they use other baths that have much higher current efficiency?

Increasing the concentration of HCl in a solution increases the number of ions and conductivity, allowing for efficient charge transfer. However, high concentrations can lead to the dominance of the hydrogen evolution reaction, decreasing the efficiency of the Zn electrodeposition process, as Reviewer mentions. Although it is possible to optimize the efficiency of Zn electrodeposition using ZnSO4 or ZnCl2, the presence of HCl only slightly inhibits charge transfer [1]. Regarding our research, the use of HCl solution for the deposition of Zn on steel allowed for the formation of a uniform and dense coating, which was visible to the naked eye. This subsequently facilitated the decrease in Open Circuit Potential (OCP) as the steel underwent dissolution. Consequently, we consider that the electrochemical sensor (steel-Zn system) developed in this research fulfills the initial requirements for our application. Nevertheless, if future research is conducted using this electrochemical sensor, we will consider using solutions with higher current efficiency for Zn deposition.

[1] J.-R. Park, H.-T. Kim, Mechanism of Zinc Electrodeposition in a Chloride Solution, (n.d.).

b) The corrosion science literature shows that for a material to undergo localized pitting corrosion, it must be in a passive state. And, as I read the manuscript, I became a little confused, as the authors at some points in the manuscript talk about localized corrosion, metastable pits, repassivation, and at others about generalized corrosion. Here are some questions: (i) Is 1010 carbon steel in the passive state in a 2 wt. % HCl solution???; (ii) For

such an explanation, it would be extremely important for the authors to resent both the potentiodynamic polarization curves and the linear polarization curves (used to determine the Rp values, in the 2 wt. % HCl solution. Given the answer presented by the authors, perhaps it would be important to review the following statement contained in the manuscript

“...It has been reported that spikes in the potential signal can be related to the occurrence of metastable pits and rapid repassivation in an electrochemical process [38] [39] [40], indicating a highly unstable electrochemical process on the steel surface when the standard

deviation value is high [41]. This, in a way, elucidates why the sample under flow conditions exhibits reduced susceptibility to repassivation and a greater propensity for the pitting process...".

Thank you very much for this observation. Authors believe that the explanation provided in the manuscript now clarifies this apparent contradictory situation:

As is well known in corrosion science, iron (and iron alloys) can be passivated under specific conditions of acidity and potential. Anodic protection has indeed been suggested as a technique to prevent corrosion in such materials and there are works proving the passivation of the carbon steel in acidic media [Mohammed, H. K., Jafar, S. A., Humadi, J. I., Sehgal, S., Saxena, K. K., Abdullah, G. H., ... & Abdullah, W. S. (2023). Investigation of carbon steel corrosion rate in different acidic environments. Materials Today: Proceedings.].

In the present case, as indicated by both the potentiostatic measurements (see figure below) and optical observations, when our samples were in contact with the acidic solution under static conditions, there were no pits, and therefore, no conditions for the generation of passive films. However, conditions changed under flow state. On the one hand, in the central part of the sample, there is a generalized type of corrosion, as in non-flow conditions. On the other hand, pitting corrosion occurs on the surface next to the borders, indicating that in such a place, there are conditions to grow a passive film.

The only reason found to explain such different behavior (in the central part of the sample and on the borders) is a different potential in both regions. As the Nernst equation and the theoretical principles of anodic protection state: by increasing the concentration of the oxidant agent, it is possible to increase the equilibrium potential of the reduction reaction, shifting the open circuit potential of the system.

In short, there is both generalized and pitting corrosion in the same sample due to the fact that the flow of the electrolyte is different in the central part than on the borders of the sample. This is because the topography of the sample on the borders is not completely flat, as it was used to fix the sample to the mechanical device.

The authors believe that the flow on such an irregular surface produces enough agitation to considerably increase the concentration of the oxidant species. For example, it has already been proven that under turbulent regimens, the oxygen concentration in the electrolyte is much larger than under laminar regimes [Chatelain, M., & Guizien, K. (2010). Modelling coupled turbulence–Dissolved oxygen dynamics near the sediment–water interface under wind waves and sea swell. Water Research, 44(5), 1361-1372.].

As the reviewer can appreciate, this is an extremely interesting topic that the authors are dealing with. However, the scope of this work cannot be extended so far. Nonetheless, it is already one of the core topics in progress in our research group. 

Figure 1. a) Potentiodynamic Polarization and b) Linear Polarization curves of 1010 carbon steel in 2 wt. % HCl solution. 

c) Before the exposure to the corrosive medium, how were the samples prepared? Were they polished, cleaned, or treated in any way? 

“Before the exposure to the corrosive medium the working area of the sample was ground with 800 grit silicon carbide paper, cleaned with soapy water, rinsed with distilled water and degreased with acetone”. This information has been added in the Experimental section. 

Surface conditions can significantly influence degradation behaviors? 

Upon the initial exposure to acid media, some observations were made on the surface of the samples. These include, i) the presence of scratches that were associated with the polishing procedure (marked with red arrows in Figure 1), and mainly ii) parallel scratches that were perpendicular to the flow on the clean, polished surface of the substrate (indicated by blue arrows). These parallel scratches were attributed to the manufacturing process (see Figure 1). Over time, the polishing scratches gradually disappeared, indicating a uniform surface dissolution during the initial stage of corrosion. This behavior is consistent with the Open Circuit Potential (OCP) signal analysis discussed in the previous question. However, tracking the manufacturing scratches over time could potentially have an impact on degradation behaviors. According to Popoola and Fayomi [1], low carbon steel showed a high susceptibility to intergranular corrosion in HCl due to the presence of various phases in the metal matrix and surface scratches. These imperfections in the steel matrix can affect the initiation and acceleration of localized corrosion in low carbon steel, as reported by Wei et al [2]. Therefore, the uniform dissolution of the steel in an acid medium begins with the activation process, as indicated by the OCP signal and potentiodynamic polarization curve. Furthermore, the influence of acid flow on the steel surface may have an impact on localized corrosion (as explained in answer to comment B) during the final dissolution stage, as depicted in Figure 7 and 8 of manuscript. In fact, the surface defects inherent to the steel, such as scratches, along with the defects caused by the optical fiber beneath the shim steel (defects due experimental set-up), could potentially influence this type of corrosion. This could be attributed to changes in shear stresses caused by both inherent and induced defects generating preferential sites for corrosion, accelerating localized dissolution. This information has been discussed in the manuscript. 

Figure 1. Tracking over time of change in morphology of 1010 carbon steel in 2 wt. % HCl solution.

[1] A.P.I. Popoola, O.S.I. Fayomi, ZnO as corrosion inhibitor for dissolution of zinc electrodeposited mild steel in varying HCl concentration, International Journal of the Physical Sciences. 6 (2011) 2447–2454. https://doi.org/10.5897/IJPS11.548.

[2] J. Wei, J.H. Dong, W. Ke, X.Y. He, Influence of Inclusions on Early Corrosion Development of Ultra-Low Carbon Bainitic Steel in NaCl Solution, Corrosion. 71 (2015) 1467–1480. https://doi.org/10.5006/1837.

d) When looking at Figure 7 of the manuscript, I get the impression that there are crevices forming at the metal/coating interface of work electrodes. Was there any special care taken to avoid this type of problem when isolating the area to be exposed to the corrosive environment? It would be important for the authors to make some mention of this and say what type of error this could lead to in the measurements.

Thank you very much for this comment. In the revised version of the article we have added a paragraph in which we discuss the possible defects generated by using glue to fix and seal the specimens. This is an issue that we will need to address in more detail in future research. 

However, it is crucial to note that no filtrations of electrolyte were detected at the metal/coating interface of the work electrodes. Otherwise, the electrochemical potential would change abruptly before the generation of pits could occur. 

The following paragraph was added to the revised version of the article:

“What is interesting to note is that the first pits were observed in areas close to the edges of the sample. As previously mentioned, cyanoacrylate was successfully used to fix the sample and prevent leakage of the electrolyte. It is worth stressing that no filtrations of electrolyte were detected at the metal/coating interface of the work electrodes. Otherwise, the electrochemical potential would change abruptly before the generation of pits could occur.However, there is the possibility of having left unevenly sized resin edges around the samples, which may affect electrolyte flow and shear stresses on the exposed metal. The lack of symmetry in these imperfections suggests non-uniform corrosion at the edges and in the area exposed to flow. Sample geometry and size may also contribute to corrosion near the boundaries. Future research will certainly need to investigate the flow velocity near the sample, using techniques such as PIV, and complement with CFD simulations to better understand the impact of geometry and edges as well as the corrosion mechanisms that give rise to pitting in these zones. Beyond 26 hours, the number of pits increased, becoming more discernible from the surrounding non-corroded areas. ”

e) When discussing Figure 5, the authors make the following comment: “...In contrast, under static conditions, the potential suffers an abrupt drop to -0.5530 V (point C). This alternating behavior persisted until point D, where the potential drops abruptly below -0.6222 V (point E) due to the appearance and growth of pits that penetrated the 25-micron thickness of the steel, exposing tiny surfaces of Zn. Subsequently, the signal converges towards potentials very close to OCP of the steel/Zn galvanic couple...”. To support this statement, why don't the authors perform potentiodynamic polarization measurements for the Zn coating and then superimpose these measurements on the steel polarization measurements? This way, the

authors could empirically prove the value of the Zn/steel galvanic couple.

While the reviewer's suggestion is well-founded, it is essential to consider the fundamental aspect of the cathode/anode area ratio in our analysis. The superimposition of both polarization measurements does not yield accurate information about the potential of the couple when a small area of the anode is in contact with a significantly larger cathodic area, akin to the scenario observed in the pitting of a steel surface with the tip of Zn. Consequently, we conducted experiments, now more comprehensively outlined in the text, wherein we measured the potential of a perforated steel sample, exposing only a minute area of Zn. The obtained potential was precisely -0.565 V, progressively decreasing down to -1.0 V as the Zn area expanded with an increasing number of perforations (once the Zn is exhausted the potential slowly shifts towards less negative values).

g) In my opinion, the conclusion must be more objective, reflecting the outcome of the work. Please be more concise and critical, highlighting the key findings. Additionally, it would be beneficial to offer suggestions for subsequent research and potential advancements in this Domain.

We agree with your assessment. In the new version of the manuscript we shortened the conclusions to make them more concise and clear. In order to enhance the suggested methods, we have highlighted a few topics that we think should be further investigated. Possible applications were also briefly mentioned.

Reviewer #3: Dear author, The paper is interesting. In my frank opinion, it deserves to be published after minor revisions:

First and foremost, thank you very much for your comments; the authors truly appreciate the time and effort invested in reviewing our work.

Minor revision of English is necessary.

A double spelling check was performed, and typos were corrected.

The images are not in good resolution.

Images 7 and 8 were modified to improve the resolution. The other figures, due to limitations inherent to the software that generates them, could not be modified to improve the resolution.

Viscosity can be changed with pressure, temperature and concentration. In formula 2, wouldn't it be necessary to take these variables into account within viscosity?

Thank you for this observation. For our experiments the fluid viscosity was kept constant as the tests were performed under constant ambient conditions (Pa=101.2 kPa and Ta=20°C). Although there was a slight evaporation of liquid during each test, a small amount of water was added to the tank to keep the HCl concentration practically unchanged. On the other hand, since the concentration of the HCl solution used is low, the viscosity of the water was considered to estimate the Reynolds number. To support this assumption, the following graph was consulted from: “Viscosities of aqueous hydrochloric acid solutions, and densities and viscosities of aqueous hydroiodic acid solutions” https://doi.org/10.1021/je00025a008

In the revised version of the article, the following lines of text were added (a reference was included along with this text):

“It is important to note that the calculation of Reynolds numbers was made considering the properties of water under constant laboratory conditions, T=20°C and P= 101.2 kPa. This does not lead to considerable errors in our estimations since the solution used for the experiments was prepared with a low concentration of HCL, so the properties of the solution are very close to those of water [a1].”

[a1] E. Nishikata, T. Ishii, and T. Ohta, Viscosities of aqueous hydrochloric acid solutions, and densities and viscosities of aqueous hydroiodic acid solutions, J. Chem. Eng. Data 1981, 26, 3, 254–256.

Why exactly use the green laser? Wouldn't it be more interesting to use a laser in the UV region? The UV laser simulates sunlight.

We find Reviewer’s proposal interesting, probably in future studies we will use a UV laser. We employed a green laser mainly because it is visible to the human eye (in contrast UV light is not visible to the naked eye) and it is brighter than other colors such as red, widely used in commercial lasers ;https://www.plslaser.com/laser-level-application-guide/why-green-lasers-brighter-than-red/. 

“The images were digitally processed using Python Programming Language to determine”. It would be interesting if you provided, as an attachment, the programming used.” 

Program 1. This program performs IMAGE segmentation using the thresholding method. Moreover, multiplies grayscale IMAGE by a manually created MASK for each exposure time. The use of a separate mask for each exposure time is necessary to compensate for mechanical device vibrations that prevent the creation of a standard mask. Consequently, a manual creation of a mask using Paint software is required for each image. The output image of this program will feed into program 2.

# -*- coding: utf-8 -*-

"""

Created on Mon Jun 26 18:37:08 2023

@author: GRECCO_5

"""

import numpy as np

import cv2

from cv2 import imread

from matplotlib import pyplot as plt

img = cv2.imread('IMAGE.bmp',0)

mascara= cv2.imread('MASK.png',0)

def segmenta(img,t):

 (N,M)=img.shape

 Y=np.zeros((N,M))

 area=0

 for i in range(N):

 for j in range(M):

 if img[i,j]>t:

 Y[i,j]=255

 area= area +1 

 print('El area es = ',area) 

 return Y

Y1=segmenta(img,200)

Y2=segmenta(mascara,150)

imgM1 = cv2.bitwise_and(Y1,Y2)

cv2.imshow('imagen',Y1)

cv2.waitKey(0)

cv2.destroyAllWindows()

cv2.imshow('imagen',imgM1)

cv2.waitKey(0)

cv2.destroyAllWindows()

Program 2. This program detects discontinuities in the segmented image (white pixels) and surrounds them with lines while counting them (which corresponds to the laser dots that indicate local dissolutions).

# -*- coding: utf-8 -*-

"""

Created on Mon Jun 26 18:37:08 2023

@author: GRECCO_5

"""

import numpy as np

import cv2

from matplotlib import pyplot as plt

img = cv2.imread(' imgM1.png',0)

cv2.imshow('imagen', img)

cv2.waitKey(0)

cv2.destroyAllWindows()

def howis(img):

 print('size = ',img.shape)

 print('max = ',np.max(img))

 print('min = ',np.min(img))

howis(img)

ret,thresh1 = cv2.threshold(img,150,255,cv2.THRESH_BINARY)

thresh1=cv2.GaussianBlur(thresh1,(5,5),0)

(contornos,jerarquia)=cv2.findContours(thresh1,cv2.RETR_EXTERNAL,cv2.CHAIN_APPROX_SIMPLE)

imagen=cv2.drawContours(thresh1, contornos,-1,(255,255,255),1)

font=cv2.FONT_HERSHEY_SIMPLEX

picadura =1

for i in contornos:

 momento =cv2.moments(i)

 cx=int(momento['m10']/momento['m00'])

 cy=int(momento['m01']/momento['m00'])

 cv2.circle(thresh1,(cx,cy),1,(255,255,255),-1)

 cv2.putText(thresh1, ""+str(picadura),(cx+5,cy+5),font,1,(255,255,255),2)

 picadura=picadura+1

cv2.imshow('imagen',imagen)

cv2.waitKey(0)

cv2.destroyAllWindows()

It is worth mentioning that the programs used for image analysis will be available to readers in an open-access repository, as required by the journal.

“the polarization resistance Rp and the Tafel extrapolation methods were utilized to determine the corrosion rate”. How the externally induced mechanical oscillation affects the charge transfer of the system: In the Tafel extrapolation, the frequency of the electric current through the material is taken into account.

Example: formula 6:

https://www.mdpi.com/2227-9717/11/3/721

There seems to be a misunderstanding, which we have clarified in the revised version of the manuscript. The Rp and Tafel extrapolation tests were performed solely under static conditions for the purpose of providing a comparative reference. As detailed in the corresponding sections, these measurements (Tafel and Rp) indicated a corrosion rate of approximately 0.7 mm/year. However, the experimental results obtained under oscillatory acidic streams demonstrated an average dissolution rate approximately twelve times faster. This significant disparity vividly underscores the aggressiveness of the conditions induced by the mechanical device described in the paper.

The study you did is very interesting and important. To emphasize the importance of this study, I suggest mentioning, for example in the introduction, other techniques for studying corrosion (currently used in static media, such as Electrochemical Impedance Spectroscopy), fuels such as electrolytes and different material manufacturing routes. Below are examples of articles that illustrate these points.

https://journals.sagepub.com/doi/abs/10.1177/00219983211029352

https://meridian.allenpress.com/corrosion/article-abstract/76/12/1109/447822/Silicon-Content-Affecting-Corrosion-Behavior-of

Thanks for the recommendation, now in the Determination of corrosion rate under activation control and no flow conditions we have added a new paragraph about EIS.

“It is important to bear in mind that, while Rp and Tafel extrapolation tests provide swift and effective corrosion rate assessments under static conditions, there exist additional electrochemical techniques such as Electrochemical Impedance Spectroscopy (EIS) that furnish valuable insights into corrosion dynamics. EIS accomplishes this by scrutinizing intricate impedance characteristics across a spectrum of frequencies, promising to unveil nuanced insights into the electrochemical dynamics of such systems. The establishment of an equivalent circuit further refines the interpretative framework, providing a comprehensive understanding of corrosion phenomena, even in dynamic environments and non-aqueous electrolytes. [GARCIA, A. et al (2009) Electrochemical corrosion response of a low carbon heat treated steel in a NaCl solution, https://meridian.allenpress.com/corrosion/article-abstract/76/12/1109/447822/Silicon-Content-Affecting-Corrosion-Behavior-of]”. 

Reviewer #4: Dear authors, below are some comments and suggestions that I consider relevant to the work and doubts that may not have been clarified in the text:

1. It is known that the central objective is not to be the result of the corrosion test but there is a focus on this area as well. It is believed to be interesting to think about applying an equivalent circuit to obtain relevant information regarding corrosion parameters. Below is an indication of reading the articles: GARCIA, A. et al (2009) Electrochemical corrosion response of a low carbon heat treated steel in a NaCl solution. and GARCIA, A. et al. (2010) Electrochemical corrosion behaviour of a Ti-IF steel and a SAE 1020 steel in a 0.5M NaCl solution., where corrosion in steel is discussed.

First and foremost, thank you very much for your comments; the authors truly appreciate the time and effort invested in reviewing our work.

Thank you for the suggestion. Indeed, we are currently incorporating EIS measurements into our ongoing research. However, the authors anticipate that the completion of this research may extend beyond spring 2024. Nevertheless, the updated manuscript now features a new paragraph addressing this specific topic.

“It is important to bear in mind that, while Rp and Tafel extrapolation tests provide swift and effective corrosion rate assessments under static conditions, there exist additional electrochemical techniques such as Electrochemical Impedance Spectroscopy (EIS) that furnish valuable insights into corrosion dynamics. EIS accomplishes this by scrutinizing intricate impedance characteristics across a spectrum of frequencies, promising to unveil nuanced insights into the electrochemical dynamics of such systems. The establishment of an equivalent circuit further refines the interpretative framework, providing a comprehensive understanding of corrosion phenomena, even in dynamic environments and non-aqueous electrolytes. [GARCIA, A. et al (2009) Electrochemical corrosion response of a low carbon heat treated steel in a NaCl solution, https://meridian.allenpress.com/corrosion/article-abstract/76/12/1109/447822/Silicon-Content-Affecting-Corrosion-Behavior-of]”. 

2. I believe the explanation for the galvanic couple formation after the first pit appearance occurs is vague. Something expected did not happen.

To answer this question, we measured the Open Circuit Potential (OCP) of the carbon steel and Zn galvanic couple in the acidic medium used in this study. Figure 1 illustrates both the black and front view of the sample used in this investigation (a), as well as the OCP of the galvanic couple between carbon steel and Zn in a 2 wt. % HCl solution (b). As expected, when this galvanic couple was exposed to an acidic medium, the OCP showed the typical potential of Zn. However, as the Zn began to dissolve, the OCP decreased in proportion to its content, causing the OCP of the galvanic couple to shift towards intermediate values. As the exposure continued over time, the OCP gradually reached a stable value, which represents the potential of carbon steel due to the complete dissolution of Zn.

In the presented investigation, the occurrence of localized dissolving in carbon steel leads to the electrolyte reaching the Zn. However, since only a small amount of Zn is reached by the electrolyte during the initial localized dissolutions (as shown in Figure 7 at 26 hours), the OCP does not decrease. As the dissolution of the steel increases, more electrolyte reaches the Zn. Due to the high activity of Zn in an acidic environment, the OCP then decreases to a more active value (as shown in Figure 5, at point D). In other words, the OCP decreases to a value between the potential of the steel and Zn (constant dissolution of small quantities of Zn), thereby consolidating the galvanic couple at a potential around -0.6222 V (dotted purple line)[1]. This information was discussed in the Results and Discussion section.

Figure 1. a) Sample of Zn coated carbon steel and b) OCP evolution of sample in 2 wt. % HCl solution.

[1] C. Arrighi, T.T. Nguyen, Y. Paint, C. Savall, L.B. Coelho, J. Creus, M.G. Olivier, Study of Ce(III) as a potential corrosion inhibitor of Zn-Fe sacrificial coatings electrodeposited on steel, Corros Sci. 200 (2022) 110249. https://doi.org/10.1016/J.CORSCI.2022.110249.

3. The analysis of the formation of pits was never discussed regarding their appearance on the edges of the samples. Is there any influence on the sample preparation or its dimensions?

Very good question, thank you. As mentioned in the article, we used cyanoacrylate to glue the samples and we do not rule out however that resin edges are left around the periphery of the specimen. Even such glue edges may not be of the same thickness (and shape) along the periphery, which of course influences the electrolyte flow, particularly by modifying the velocity field and the shear stresses on the exposed metal. Obviously, the possibility that there is no symmetry of these imperfections along the periphery of the specimen also implies that the corrosive effect is also not uniform at the edges and even over the entire area of metal exposed to the flow. The geometry and size of the samples can also play an important role in the pitting near the edges because of the aspects mentioned above. In future research, it would be necessary to perform flow velocity measurements in areas close to the sample (using for example the PIV technique) and complement them with CFD simulations to clarify the effect of sample geometry and edges, however this is beyond the scope of the present work.

In view of the above, the following lines were added to the revised version of the manuscript:

“What is interesting to note is that the first pits were observed in areas close to the edges of the sample. As previously mentioned, cyanoacrylate was successfully used to fix the sample and prevent leakage of the electrolyte. It is worth stressing that no filtrations of electrolyte were detected at the metal/coating interface of the work electrodes. Otherwise, the electrochemical potential would change abruptly before the generation of pits could occur. However, there is the possibility of having left unevenly sized resin edges around the samples, which may affect electrolyte flow and shear stresses on the exposed metal. The lack of symmetry in these imperfections suggests non-uniform corrosion at the edges and in the area exposed to flow. Sample geometry and size may also contribute to corrosion near the boundaries. Future research will certainly need to investigate the flow velocity near the sample, using techniques such as PIV, and complement with CFD simulations to better understand the impact of geometry and edges as well as the corrosion mechanisms that give rise to pitting in these zones. Beyond 26 hours, the number of pits increased, becoming more discernible from the surrounding non-corroded areas.”

---

## [Editor Report · Decision Letter 1]

23 Jan 2024

Accelerated corrosion of low carbon steel by oscillatory acidic streams generated with a bio-inspired claw device

PONE-D-23-36475R1

Dear Dr. Montoya,

We’re pleased to inform you that your manuscript has been judged scientifically suitable for publication and will be formally accepted for publication once it meets all outstanding technical requirements.

Kind regards,

Wislei Riuper Osório

Academic Editor

PLOS ONE

Additional Editor Comments (optional):

Based on the Reviewers’ comments and the rebuttals provided by Author, it is observed that the manuscript was meticulously revised and considerably improved. With this, it deserves its final publication.
---

## [Editor Report · Acceptance letter]

23 Mar 2024

PONE-D-23-36475R1 

PLOS ONE

Dear Dr. Montoya, 

I'm pleased to inform you that your manuscript has been deemed suitable for publication in PLOS ONE. Congratulations! Your manuscript is now being handed over to our production team.

Kind regards, 

on behalf of

Dr. Wislei Riuper Osório 

Academic Editor

PLOS ONE